# The Anticancer Drug Discovery Potential of Marine Invertebrates from Russian Pacific

**DOI:** 10.3390/md17080474

**Published:** 2019-08-16

**Authors:** Vladimir L. Katanaev, Salvatore Di Falco, Yuri Khotimchenko

**Affiliations:** 1Department of Cell Physiology and Metabolism, Translational Research Centre in Oncohaematology, Faculty of Medicine, University of Geneva, Rue Michel-Servet 1, 1211 Geneva, Switzerland; 2School of Biomedicine, Far Eastern Federal University, 8 ul. Sukhanova, 690950 Vladivostok, Russia; 3The Institute of Economics and Econometrics, University of Geneva, UNIMAIL, Boulevard du Pont d’Arve 40, 1211 Geneva, Switzerland; 4National Scientific Center for Marine Biology, Far Eastern Branch of Russian Academy of Sciences, 690041 Vladivostok, Russia

**Keywords:** marine natural products, cancer, drug discovery, North Pacific, Russia, invertebrates, biodiversity, chemodiversity, pharmacophore diversity, national strategic initiative

## Abstract

Despite huge efforts by academia and pharmaceutical industry, cancer remains the second cause of disease-related death in developed countries. Novel sources and principles of anticancer drug discovery are in urgent demand. Marine-derived natural products represent a largely untapped source of future drug candidates. This review focuses on the anticancer drug discovery potential of marine invertebrates from the North-West Pacific. The issues of biodiversity, chemodiversity, and the anticancer pharmacophore diversity this region hides are consecutively discussed. These three levels of diversity are analyzed from the point of view of the already discovered compounds, as well as from the assessment of the overall, still undiscovered and enormous potential. We further go into the predictions of the economic and societal benefits the full-scale exploration of this potential offers, and suggest strategic measures to be taken on the national level in order to unleash such full-scale exploration. The transversal and multi-discipline approach we attempt to build for the case of marine invertebrate-based anticancer drug discovery from a given region can be applied to other regions and disease conditions, as well as up-scaled to global dimensions.

## 1. Introduction

Possessing an unprecedented molecular diversity, natural products have been the source of numerous medicinal drugs and health benefit products throughout human history, from traditional medicines to modern molecular drug discovery approaches [1]. Today, of all marketed drugs, about 40% originate from natural compounds (either directly or through synthetic modifications), and this share grows to ca. 50% when looking at small molecule drugs [2]. In this regard, medicinal plants have been used in traditional medicines for thousands of years and have so far been the focus of modern drug discovery research. On the other hand, marine-derived drugs represent a minority, as their exploitation started only in the 1970s, but are of a high promise, as the chemical novelty and diversity of marine-derived natural products exceeds that of terrestrial sources [3]. The biggest (and alternative to the terrestrial plants or organic synthesis laboratories) chemical diversity marine organisms offer is multiplied by the biggest biodiversity of the marine life forms on our planet [4,5]. Although the marine reserve of the natural treasury of biologically active compounds has only recently begun to be explored, it has already led to the introduction of a number of important drugs into the clinic.

As compared to the terrestrial natural products, the chemical and pharmacophore diversity of marine natural compounds has remained mostly untapped. Yet it is certain that continuing exploration of the biodiversity of marine environments will inevitably bring more drugs originating from marine compounds. The development of marine-based drugs is expected to grow exponentially. The biggest share of those new drugs will target various forms of cancer, as do the pioneers in the marine natural product-based drug discovery, the anticancer medicines Erubilin mesylate, Trabectedin, or Brentuximab vedotin [6].

The current review focuses on invertebrate marine natural products with an anticancer potential. However, instead of listing the currently known compounds as several recent articles have done in detail [7,8,9], we try to present a different angle of analysis. Focusing on one, yet huge, marine region (North-West Pacific), we will aim at providing four different types of cross-section of the anticancer potential of this region. The first will overview the dynamics of the discovery of marine invertebrates in this region, permitting us to reevaluate the overall estimate of the invertebrate biodiversity of the North-West Pacific. The second will similarly probe the overall chemodiversity, which this biodiversity has to offer. The third will cut through the chemodiversity subspace with the anticancer potential. The fourth will attempt to evaluate the commercial and societal value, in terms of the drugs to be developed and the health conditions to treat, of the full-scale exploration of the said anticancer chemodiversity of the ‘treasure chest’ hidden in the North-West Pacific.

To our knowledge, such an investigation combining biological, chemical, and economic assessments has not previously been performed to study the drug discovery potential of marine natural products. Being restricted to a given marine region of the North-West Pacific, this approach can later be applied to other domains and upscaled to encompass the global Ocean. We also hope that this analysis may be beneficial for the governmental strategic planning.

## 2. Assessing the Biodiversity of Marine Invertebrates of the North-West Pacific

The biodiversity of the Russian Pacific has been a focus of regular expeditions organized by the Russian Academy of Sciences starting from the 1960s until currently. Three works summarize the results of these massive investigations, revealing the gross dynamics of discovery of the new species: Biology of the seas of the USSR, 1963, by L.A. Zenkevich [10]; biological diversity of invertebrates in Far Eastern seas of Russia, 1994, by B.I. Sirenko [11]; and check-list of species of free-living invertebrates of the Russian Far Eastern seas, 2013, edited by B.I. Sirenko [12]. These works reveal the steady rate of accretion of new invertebrate species in the North-West Pacific, from ca. 3000 in 1963 to ca. 8500 in 2013 (Table 1).

According to [12], invertebrates of the Russian Pacific consist of the following three major groups: Macrobenthos (comprising ca. 63% of the currently known species), meiobenthos (25%), and plankton (12%). Of different invertebrate groups, currently the most divergent in the North-West Pacific are crustaceans with 3190 described species (39% of all invertebrates). They are followed by mollusks (1348 species or 17% of all invertebrates), annelids (800 species, 9.5%), and cnidarians (406 species or 4.8%). Protists, mainly Granuloreticulosa, Rhizaria, Cercozoa, and Ciliophora are represented by 731 known species (8.7%).

The least studied marine invertebrate group is the meiobenthos, which includes bottom-dwelling animals with the small body size from 0.1 to 2 mm. Given the apparent incompletion of the discovery of the small meiobenthic species, belonging to the groups of Nematoda, Plathelminthes, Tardigrada, Harpacticoida, Arachnida, and Ostracoda, it is estimated that the number of yet to be discovered meiobenthic species in the North-West Pacific is >10,000 [12].

It is apparent that the number of undiscovered species increases with the depth of the habitat. To close this gap, A.V. Zhirmunsky National Scientific Centre of Marine Biology (Far Eastern Branch of Russian Academy of Sciences) performed, in 2010–2016, a series of expeditions investigating marine biodiversity at big depths of the Russian Far Eastern seas, as well as the Kuril–Kamchatka trench in the North Pacific.

In 2010, the Russian–German expedition SoJaBio-2010 on the research vessel “Academician Lavrentyev” collected samples along the big depth trans-section from 500 to 3660 m in the Sea of Japan. This expedition collected 621 invertebrate species, of which 201 were previously unknown, and 105 were unknown for the Sea of Japan [13].

In 2012, the KuramBio I expedition investigated the abyssal plate adjacent to the Kuril–Kamchatka trench. Of the specimen collected along the trans-section of 4830–5830 m, 1780 invertebrate species were found, of which a half (869 species) were previously unknown [14].

During the expedition SokhoBio-2015 at the deep Kuril basin of the Sea of Okhotsk, >1000 invertebrate species were collected along the trans-section of 1700 m to 4700 m, of which ca. 50% were previously unknown [15,16].

Hadal depths were also discovered to contain an unexpected degree of biodiversity. The Russian–German expedition KuramBio II, investigating the Kuril–Kamchatka trench at the depths down to 9500 m in 2016, collected >1300 invertebrate species, of which, again, about a half represented species totally unknown to science previously [16].

We would like to make four important conclusions to this part of our review. First, the biodiversity of the North-West or Russian Pacific, already discovered to date, by far exceeds the estimations that could have been based on the latitudinal biodiversity gradient, characterized by the decrease in species richness from equator to poles [17]. While the marine biomass clearly follows the north-to-south gradient [18], the species diversity, as observed e.g., in the recent pan-ocean analysis of ophiuroids [19] and marine viruses [20], may not necessarily follow the same trend. In other words, northern altitudes may hide no less biodiversity than the tropical ones.

Second, early expectations governing among biologists before the launch of the deep-sea exploration were that the biodiversity decreases with the depth of the ocean, from the benthal to abyssal to the hadal depths. However, modern deep-sea research has since long challenged this assumption [21], and the recent studies e.g., on the biodiversity of ophiuroids [19,22] provide direct data against it, and the recent studies in the North-West Pacific agree well with these analyses [16]. With the peculiar ecosystems hadal environments can hide, such as hot vents, ocean depths actually appear to host rich and unique species diversities [23].

The third and most important for our review conclusion is the ultimate estimate of the invertebrate biodiversity the North-West Pacific contains. With the data from the deep-sea expeditions of 2010–2016 added to the prior analysis, we arrive at ca. 8411 (Sirenko, 2013 [12]) + 201 (SoJaBio-2010 [13]) + 869 (KuramBio l [14]) + ca. 500 (SokhoBio-2015 [15,16]) + ca. 650 (KuramBio II [16]) = ca. 10,630 described species. This number already is very impressive. We would like to stress, however, that in the recent series of expeditions, each cruise brings about roughly as many new species as the species previously described, and this remarkable pace does not seem to fade. To repeat that abyssal and meiobenthic invertebrates remain to a large extent undiscovered, we would like to provide the cautious estimate that the overall invertebrate biodiversity of the North-West Pacific exceeds 20,000 species, and the optimistic estimate that this number approaches 100,000 species. These numbers correlate with the estimate of Snelgrove that at least 50%, but as much as 90% of marine species, on the global scale, remain undiscovered [4]. We would like to add to that a daring estimate of Adrianov [5] on the overall number of deep-sea macrobenthos and meiobenthos: With his estimate of ca. 50 million species globally of these types of invertebrates remaining to be discovered, the overall invertebrate biodiversity of the North-West Pacific can reach up to 2 million species.

The fourth note we would like to make refers to the taxonomic biodiversity of marine invertebrates. Species number-wise, the optimistic estimate of the North-West Pacific invertebrates is far lower than that of terrestrial invertebrates (of which the biggest diversity is provided by insects, with ca. 1 million described insect species). However, the taxonomic diversity of marine invertebrates is much bigger than of their terrestrial cousins. Indeed, of the existing metazoan phyla, only 11 are represented above the water, while the marine invertebrates comprise 31 phyla [5]. This taxonomic diversity of marine invertebrates, with the 20-to-100-to-2000 thousand species diversity estimated for the North-West Pacific region only, provides an enormous treasury of the biological and biochemical nature, largely untapped in terms of technological developments. The next section will speak about the conversion of this biodiversity to the underlying chemodiversity of North-West Pacific invertebrates.

## 3. Natural Products from North-West Pacific Invertebrates

Globally, with some 180,000 marine invertebrates currently known, about 30,000 marine-derived natural products have been described, and this number increases by ca. 1000 annually [24]. An analysis of marine-based natural products from the perspective of their pharmacophore diversity and drug likeliness, in comparison with the terrestrial-derived natural products and similarly sized compounds obtained through combinatorial organic synthesis, has been performed [25]. Among the three groups, the marine-based natural products not only showed the highest chemical diversity in terms of structures, but were also found to possess the highest proportion of drug-like compounds. This computational analysis is corroborated by the drug discovery and development pipeline stemming from marine compounds. With ‘only’ ca. 30,000 marine-based compounds so far described (a dwarf number as compared to the chemically synthesized and terrestrial-based natural products), >40 have already made it to clinical trials [26], representing the drug discovery success rate of an unparalleled power. These numbers already represent the ‘seeds’ of the enormous commercial and health-related potential of the anticancer compounds of marine invertebrates we elaborate on in the next sections. Here, we would like to spare some words on the potential reasons behind these high numbers. Following [25,27], we speculate that the number of potential protein folds (translated into the number of pharmacophore-binding sites in drug targets) is limited, and secondary metabolites, having co-evolved in living organisms to interact with certain protein folds, have a high chance to find a new match when probed against human disease-related protein targets [25,28]. This would be an argument for the superiority, in terms of the drug likeliness, of natural products over those amenable to organic synthesis. After all, about a half of all currently marketed drugs originate, directly or through chemical modifications, from natural products [2,24]. The apparent superiority of marine-based natural products over the terrestrial (mainly plant-derived) compounds may be related to the higher chemical diversity of marine compounds [25], which in turn might stem from the higher taxonomic diversity hosted by marine habitats [5] as well as from the higher diversity of environments in which the organisms live, including the harsh conditions of big depths [4,25].

Alejandro Mayer from Midwestern University maintains a highly informative website tracking clinical development of marine-derived drugs [26]. Currently, this website lists eight such drugs as approved, six in phase III, 14 in phase II, and 10 more in phase I clinical trials. Of this total of 38 molecules, 32 are anticancer agents, mostly of the cytotoxic or cytostatic type. This list can be further expanded somewhat. Indeed, in addition to omega-3 acid ethyl esters (Lovaza^®^), icosapent ethyl (Vascepa^®^) and omega-3 carboxylic acids (Epanova^®^) have been approved [29]; keyhole limpet hemocyanin is used as an anticancer vaccine conjugate [30]; linear sulfated polysaccharides from Rhodophyceae seaweeds are applied as an anti-viral agent [31]; or OligoG derived from marine algae is being tested for cystic fibrosis in phase II trials [32]. The oldest marine-derived drug is cytarabine, which is the synthetic analog of sponge-isolated cytostatics spongothymidine and spongouridine. Cytarabine was approved as an antileukemic agent in 1969. Other marine-based anticancer medicines are Erubilin mesylate, Trabectedin, and Brentuximab vedotin [6]. An eEF1A2-targeting peptide from tunicates, Plitidepsin, has been recently added to this list [26].

Russia, with its >40,000 km-long coastal line of 12 seas belonging to three oceans, its extensive marine fleet with the centuries-old tradition of marine expeditions around the globe, and its solid scientific tradition of marine research is ideally positioned to become an international leader in the quest to bring marine natural products to the service of mankind. Research on marine natural products is conducted in several centers in Russia: N.N. Vorozhtsov Novosibirsk Institute of Organic Chemistry, Ufa Institute of Chemistry, N.D. Zelinsky Institute of Organic Chemistry (Moscow)*,* A.V. Zhirmunsky National Scientific Center of Marine Biology (Vladivostok), and G.B. Elyakov Pacific Institute of Bioorganic Chemistry (Vladivostok), the latter being the most important in characterization of marine natural compounds from the Russian Pacific.

Among the marine natural compounds described by G.B. Elyakov Pacific Institute of Bioorganic Chemistry are alkaloids, steroids, glycosides, cerebrosides, terpenoids, chinoid metabolites, polyphenols, oligo- and polysaccharides, lipids, peptides, and other types of compounds. As an example of a marine-based compound in clinical development from Elyakov’s Pacific Institute of Bioorganic Chemistry, one can mention cumaside, a drug candidate derived from a holoturian *Cucumaria frondosa japonica*. Cumaside possesses an immunostimulatory capacity, decreases the multidrug resistance of cancer cells, and further reveals a radio-protective activity [33].

G.B. Elyakov Pacific Institute of Bioorganic Chemistry describes ca. 50–100 new marine compounds annually, with the overall toll of such compounds exceeding 500 [34], by far pioneering any other scientific center in the country. This pace of discovery of new marine-derived compounds correlates with the global trend: Of the total 28,609 currently described marine-derived natural products, the annual rate of discovery of new compounds has grown from 332 in 1984 to ca. 1300 in the recent years (1378 new marine compounds in 2014, 1340 in 2015, and 1277 in 2016) [24,35,36]. It is clear that the overall chemodiversity marine organisms possess correlates with the biodiversity of the marine habitats. The nature of this correlation is unlikely to be linear, though. On one hand, only a minority of the currently described 180,000 marine invertebrates have been studied in terms of their metabolite composition. Continued studies on the metabolomes of the known invertebrates will strongly increase the current score of ca. 30,000 marine-based natural products described. In this regard, a general assessment of how many metabolites a given organism can possess (the human metabolome, for example, contains more than 100,000 compounds of the abundance from nM to µM range [37]) gives an approximation of what may be expected from the already known marine invertebrates. On the other hand, it is clear that we are interested in novel compounds, while a large proportion of the metabolites is shared among different species, especially among those related to each other [38]. The high taxonomic and ecological diversities marine environments provide [3,5] give in this regard a reason to expect higher chemical diversities than e.g., from terrestrial invertebrates.

With these considerations providing only a vague framework for the build-up of estimates of the chemodiversity, we assess the factor by which the number of invertebrate species should be multiplied in order to obtain the number of the resultant unique chemical species to lie anywhere between 0.1 and 10. Laid upon the estimate of the invertebrate biodiversity in the North-West Pacific as provided in the previous section, this factor leads to the calculation of the resultant chemodiversity as varying from 2000 up to 20 million unique pharmacophores. The lower bound here is clearly a vast underestimation, as it is close to the number already discovered in G.B. Elyakov Pacific Institute of Bioorganic Chemistry alone [34]. Within this broad range, our ‘intelligent guess’ is that 100,000 to 1,000,000 unique compounds are to be offered by invertebrates of the North-West Pacific. This number, especially when further amplified by the available methods of synthetic diversification, chemical modification, or enzymatic modification (see [39] as a recent example of the later), represents an enormously rich source of potential new drugs. How many of them could have an anticancer potential is addressed in the next section.

## 4. Marine-Derived Compounds from the North-West Pacific Invertebrates with an Anticancer Potential

Currently, of the >500 marine invertebrate natural products described from Russian Pacific, ca. 60 compounds have been attributed to possess anticancer activities in preclinical settings [34]. We list some of them below by classes, briefly specifying their characteristics. The sources of the natural products listed below are provided in Table 2, and some examples of these compounds in Table 3.

### 4.1. Alkaloids

Extracts from the sponge *Monanchora pulchra*, collected during the expeditions of the vessel “Academician Oparin” in 2008 in the vicinity of the Urup island of the Kurils archipelago at the depth of ca. 100 m, were found to be cytotoxic against the human acute monocytic leukemia cell line THP-1. This prompted isolation of the monanchocidin A as the active component—a guanidine alkaloid with an unprecedented skeleton system derived from a polyketide precursor, (ω-3)-hydroxy fatty acid, and containing a 2-aminoethyl- and 3-aminopropyl-substituted morpholine hemiketal ring [53]. At micromolar concentrations, monanchocidin A demonstrated cytotoxicity against human leukemia THP-1 (IC_50_ 5 μM), human cervix epithelioid carcinoma HeLa (IC_50_ 12 μM), and mouse epidermal JB6 Cl41 (IC_50_ 12 μM) cell lines [53]. Subsequent studies revealed sub-micromolar IC_50_s for cytotoxicity against a panel of cell lines form genitourinary, prostate, and bladder cancers [54]. Monanchocidin A was found to be synergistic with cisplatin in vitro, and induction of autophagy was highlighted as the mode of cytotoxic action of the alkaloid [54]. Monanchocidin represents a family of compounds comprised of monanchocidin A, monanchocidin B, monanchomycalin C, ptilomycalin A, monanchomycalin B, normonanchocidin D, urupocidin A, and pulchranin A. Most of them were shown potent in causing cell cycle arrest and apoptosis in HeLa and JB6 P+ Cl41 cancer cells at low micromolar concentrations. The compounds were further shown to inhibit the EGF-induced neoplastic transformation of JB6 P+ Cl41 cells [55].

### 4.2. Glycosides (Saponins)

Glycosides (saponins) represent a large group of marine bioactive compounds. These natural glycosides contain either a triterpene or a steroid aglycone and are typically found in the Asteroidea and Holothuroidea classes of Echinodermata. Of those, the starfish species mostly possess steroid saponins further divided into asterosaponins, cyclic glycosides, and polyhydroxysteroid glycosides, while triterpene saponins containing an oligosaccharide chain and an aglycone based on holostan-3β-ol are more typical for holothurians.

### 4.3. Triterpene Glycosides

The triterpene glycoside frondoside A was first isolated from the edible sea cucumber *Cucumaria frondosa japonica*, which is common to North Atlantic and North Pacific [59], and was subsequently also isolated from its relative from the Sea of Okhotsk, *Cucumaria okhotensis* [46]. Frondoside A demonstrated effects against human prostate cancer cells, both in vitro and in vivo [60]. In a panel of five human prostate cancer cell lines, frondoside A inhibited cell proliferation in vitro with low micromolar and sub-micromolar IC_50_s; colony formation by the cancer cells was also efficiently suppressed. Cell cycle arrest and apoptosis were behind the effect of the compound; the capacity of the glycoside to inhibit late stages of autophagy (which plays cytoprotective function in prostate cancer cells) was also found [60]. To evaluate the effects of frondoside A in vivo, mouse xenograft models using subcutaneously transplanted human prostate cancer cell lines PC-3 and DU145 were used. Remarkably, in both models frondoside A could suppress the primary tumor growth, lung metastasis, and the numbers of circulating tumor cells without strong side effects. Elevated lymphocyte counts suggested to the authors an immune modulating response as a potential mechanism, in addition to the primary suppression of the cancer cell proliferation, of the in vivo anti-tumor effects of the compound. These findings highlight frondoside A as a promising drug candidate for prostate cancer treatment [60]. Promising effects of the glycoside were observed in vitro also against other types of cells. Anti-proliferative effects were described in EGF-driven mouse epidermal JB6 Cl41 P^+^ cells, human cervix carcinoma HeLa and leukemia HL-60 cells [61]. Potential effects on the multi-drug resistance, through blockade of the membrane transport by P-glycoprotein (also known as MDR1), were seen in ascites of mouse Ehrlich carcinoma cells [62]. Frondoside A at low micromolar concentrations also showed high cytotoxicity through apoptosis induction in a panel of human urothelial carcinoma cell lines; it also demonstrated synergism with cisplatin and gemcitabine [63].

A compound similar to frondoside A, cucumarioside A2-2 from *C. f. japonica* collected from Peter the Great Bay, Sea of Japan, was also shown to block the membrane transport by P-glycoprotein in ascites of mouse Ehrlich carcinoma cells [62], and to suppress colony formation and proliferation of EGF-driven mouse epidermal JB6 Cl41 P^+^ cells and of human cervix carcinoma HeLa cells at low micromolar concentrations [61].

Cumaside—a drug candidate derived from *C. f. japonica* mentioned above—represents a mixture of monosulfated triterpene glycosides (mainly cucumarioside A2-2) and cholesterol at the molar ratio of ca. 1:2 [64]. Unlike its constituent glycosides, Cumaside shows reduced cytotoxic and hemolytic activities, but instead reveals immunomodulatory properties [33,65]. Interestingly, this immunostimulatory activity of cumaside is apparently behind its in vivo antitumor effect against mouse Ehrlich carcinoma upon oral application [33].

The Far Eastern sea cucumber *Eupentacta fraudatrix* collected from Peter the Great Bay, Sea of Japan, was the source of cucumariosides A1, A3–A6, A12, A15, and D1, of which A1, A3, and A5 were shown to possess high cytotoxic activity against mouse spleen lymphocytes and Ehrlich carcinoma cells [47,48].

Another triterpene glycoside from the edible sea cucumber *Apostichopus japonicus* collected from Posiet Bay, Sea of Japan, holotoxin A1 induces apoptosis in human leukemic and colorectal cancer cells at sub-micromolar concentrations (IC_50_ ranging from 20nM to 1 μM depending on the cell line) [50,51]. Holotoxin A1 was found to induce apoptosis through activation of acid sphingomyelinase and neutral sphingomyelinase. The compound was also highly potent against a set of primary patient leukemic cells, but not normal hematopoietic progenitor cells, highlighting its potential for further drug development [51].

The sea cucumber *Psolus fabricii* collected in the sublittoral of the island of Onekotan, Sea of Okhotsk [49], was the source of multiple triterpene glycosides including psolusosides A, B, C_1_, C_2_, and D_1_ [49,66,67,68]. Of those, the inhibitory activity of psolusoside А was seen at sub-micromolar concentrations using EGF-driven mouse epidermal JB6 Cl41 P^+^ cells in colony formation and proliferation assays [61]. More recent studies have described isolation of additional sulfated triterpene glycosides from the same species, psolusosides B, E, F, G, H, H_1_, and I, of which psolusoside E demonstrated low micromolar IC_50_ in inhibition of proliferation of mouse neuroblasoma Neuro 2A cells, and nanomolar IC_50_ in inhibition of colony formation of human colorectal adenocarcinoma HT-29 [69].

Sulfated triterpene glycoside fallaxoside D_1_ from *Cucumaria fallax* showed cytotoxic activity against neuroblastoma Neuro 2A cells with IC_50_ of 38 μM [44].

### 4.4. Steroid Glycosides

The Far Eastern starfish *Leptasterias ochotensis* (order Forcipulatida, family Asteriidae) was collected by the 29th expedition of the research vessel “Academician Oparin” in August 2003. The specimen collected at depths of 20–40 m in the Sea of Okhotsk near the island of Bolshoy Shantar was the source of six new polar steroids (asterosaponins), leptasteriosides A–F. These compounds showed varying activities in growth inhibition and colony formation in vitro on human breast cancer T-47D and melanoma RPMI-7951 cell lines. In all assays, leptasterioside B was the strongest, with IC_50_ in growth inhibition of 2 μM (T-47D cells) and nearly compete prevention of colony formation of same cells at 10 μM, appearing as a promising candidate for further anticancer development [41]. *L. ochotensis* was also the source of three new sulfated steroid monoglycosides, leptaochotensosides A–C and a new sulfated polyhydroxylated steroid. Of these new steroids, leptaochotensoside A inhibited colony formation (but not proliferation) of breast cancer T-47D cells with IC_50_ of ca. 50 μM; even higher concentrations were required to inhibit EGF-induced colony formation and MAPK signaling in mouse epidermal JB6 Cl41 cells [42].

The starfish *Lethasterias fusca* collected in Posyet Bay in the Sea of Japan at a depth of 5–10 m was the source of two new asterosaponins, lethasteriosides A and B, alongside with the previously identified from other sources thornasteroside A, anasteroside A, and luidiaquinoside. Of these compounds, lethasterioside A produced nearly complete suppression of colony formation of human breast T-47D, colorectal carcinoma HCT-116, and melanoma RPMI-7951 at 20 μM, although it was non-cytotoxic against T-47D and exhibited limited cytotoxicity against HCT-116 (IC_50_ 40 μM) and RPMI-7951 (IC_50_ 30 μM) cells. Based on these observations, lethasterioside A emerges as a promising candidate for further investigation as an anticancer agent [40].

Another Far Eastern starfish *Hippasteria phrygiana*, collected in the Sea of Okhotsk near Kuril Islands, was the source of four new sulfated steroidal glycosides (asterosaponins), hippasteriosides A–D, characterized by an unusual hexasaccharide moiety, β-d-xylopyranosyl-(1→3)-β-d-fucopyranosyl-(1→2)-β-d-quinovopyranosyl-(1→4)-[β-d-quinovopyranosyl-(1→2)]-β-d-xylopyranosyl-(1→3)-β-d-quinovopyranosyl, linked to C(6) of 3-*O*-sulfonylated steroidal aglycons. Of the four compounds, hippasterioside D at 60 μg/mL strongly suppressed the size of colonies of HT-29 colon cancer cells, while colony numbers were only moderately suppressed. Cytotoxicity against these cancer cells was also not observed [43].

### 4.5. Polyketides

Ethanol extract of the ascidian *Polysincraton* sp. collected at the depth of ca. 200 m near Urup island, Sea of Okhotsk, revealed strong cytotoxicity against human cervix cancer HeLa cells and was processed by fractionation and characterization of the active compound, which turned out to be mycalamide A—previously described from a New Zealand sponge [70]. The *Polysincraton* sp.-purified mycalamide A inhibited EGF-induced neoplastic transformation in murine JB6 Cl41 P+ cells at remarkable subnanomolar concentrations (IC_50_ of 0.1nM); apoptosis-mediated cytotoxicity was also observed in these cells with the IC_50_ of 6nM [58]. Similar effects on colony formation and cytotoxicity were observed for HeLa cells. Low nanomolar concentrations of mycalamide A suppressed AP-1- and NF-κB-dependent oncogenic transcriptional activities in JB6 Cl41 cells. The authors conclude that mycalamide A shows promising potential for both cancer-prevention and cytotoxic therapy and should be further developed [58].

### 4.6. Fatty Acids

The Far Eastern marine sponge *Melonanchora kobjakovae* collected at the depth of 120 m near Urup island, Sea of Okhotsk, was the source of melonoside A, the first representative of a new class of ω-glycosylated fatty acid amides. Melonoside A (10 μM) was found to induce autophagy of human cisplatin-resistant germinal tumor cells NCCIT-R [56]. Related compounds, melonoside B and melonosins A and B were later identified from the same sponge upon additional sample collections. Melonosin A was found to inhibit AP-1- and NF-kB-dependent transcriptional activities in JB6 Cl41 cells with the IC_50_ of ca. 7 μM [57].

This description provides some illustrations of the natural compounds from the North-West Pacific invertebrates with certain anticancer properties. With a few exceptions, such properties were studied only in in vitro settings, highlighting a long path these compounds have to go before they can be considered real anticancer drug candidates. New levels of experimentation have to be added to the currently existing pipeline of marine-based drug discovery of the North-West Pacific compounds, as will be elaborated in later sections. The overall number of compounds with certain anticancer effects isolated from the Russian Pacific (around 60, [34]) is lagging behind the global score of ca. 1700 compounds with certain anticancer properties described [8]. With the enormous chemodiversity the North-West Pacific offers (see above), intensification of marine-based drug discovery in this region is expected to produce a boom in the discovery of new anticancer compounds. Based on the global data, ca. 6% of marine-based natural products are expected to possess anticancer properties (ca. 1700 compounds of the total 28,609 compounds [8,35]). This gives a rough estimate that the North-West Pacific, with its 100,000 to 1,000,000 unique compounds only from the invertebrates, offers 6000–60,000 chemical species with anticancer properties. How many of those may be expected to reach the early phases of clinical trials (which is also a milestone for commercialization) will be considered in the next session.

## 5. The Drug Discovery Potential of Invertebrate Compounds of the North-West Pacific

To reiterate the global data mentioned several times above, ca. 30,000 marine-based compounds have been described, of which ca. 2000 have been ascribed certain medically relevant characteristics (with ca. 1700 showing anticancer activities in preclinical settings) [8,35]. These numbers have translated into a dozen of approved drugs and >30 more in different phases of clinical trials; the majority of these molecules are anticancer agents [26,29,30,31,32].

Compared to the terrestrial-based natural products and especially, to the chemically synthesized small molecules, this statistics represents the drug discovery success rate of an unparalleled power [71]. In general, the drug discovery/drug development process consists of several well-established steps, such as target identification and validation, hit identification, hit-to-lead optimization, animal studies (all these refer to preclinical steps), phase I-II-III clinical studies, approval, and post-market launch activities. On average, 90% of drug candidates are lost at each consecutive step prior to drug approval [71]. Clearly, marine-based natural products reveal much lower failure rates than other groups of drug candidates, with ca. 2.5% of the compounds showing preclinical anticancer activities reaching at least phase I of the clinical studies.

In the North-West Pacific, only ca. 500 compounds have been discovered; only ca. 60 of them are described to possess anticancer properties in vitro; a few have been studied in the animal settings; and only one—cumaside—has made it to the level of clinical trials and commercialization. This contrasts with the global data and especially—with the estimates of the potential of this region. As discussed above, the invertebrates of this region alone promise to provide 100,000 to 1,000,000 unique compounds, 6000–60,000 of which expected to have anticancer properties in preclinical settings. Based on the global data, no less than 150-to-1500 of these compounds should be able to reach the clinical drug development stages. From the perspectives of the national pharmaceutical industry, these are enormous numbers, especially considering that most of them will be novel, first-in-class molecules, as opposed to the “me-too” drug candidates dominating some of national pharmaceutical businesses [72]. We will now try to estimate the economic and societal importance of these numbers.

The global oncology drugs market was valued at $96 billion in 2013, reached $133 billion in 2017, and is projected to reach $200 billion by 2022 [73]. Both today and by 2025, targeted therapies (see below) take the biggest share in the global oncology market, 2–3 times bigger than classical chemotherapies and immunotherapies. Breast, prostate, kidney, and gastric cancers are among the leading indications in this regard [74].

The number of novel therapies for cancer released on the market in the USA is ca. 15 a year (14 in 2017 [73], 16 in 2016 [75]). 700–800, of which about half are small molecules, anticancer drug candidates are currently at different stages of clinical development; ca. 80% are estimated to be potential first-in-class molecules. These numbers indicate that the projected number of 150-to-1500 of marine invertebrate-based compounds from the Russian Pacific might be capable of feeding the market comparable to the oncology drug market of the USA, which takes the biggest share of the current global value. This rough approximation highlights the enormous economic potential of the marine compounds of the North-West Pacific.

From the societal point of view, in should be noted that cancer is the second leading cause of death in developed countries (after cardiovascular diseases) and the third—in developing countries (after infectious and cardiovascular diseases). According to the World Health Organization, close to 10 million people have died from cancer in 2018 on the global scale. These numbers will be growing with the overall population rise, and especially with the growth in the proportions of aged people. The global economic loss due to cancer is above $1 trillion annually [76] ($8.1 billion annually in Russia [77]). Despite the enormous efforts and numerous achievements in the prevention and treatment of cancer, the overall improvements as measured e.g., by the global mortality are modest. Indeed, comparison of the period of 2006–2010 to the period of 1981–1985 shows a 17% decrease in the cancer-related mortality in countries with very high human development index (HDI), and a mere 5% decrease in the countries with medium-high HDI. This is contrasted by the impressive success in the combat against cardiovascular diseases: a 53% drop of mortality in very high HDI countries and a 19% drop in the countries with medium-high HDI [78]. It is thus obvious that new drugs to treat and to prevent cancer must be searched for. In this regard, natural compounds, such as those derived from marine invertebrates, especially if they represent novel targeted therapies instead of being general cytotoxics (see the next section), may emerge as a novel trend to overcome the dire situation with the cancer-related mortality.

Finally, we would like to touch upon the issue of biopreservation. In general, diversity is crucial for the production of a range of marketed goods and services [79]. One important economic value of diversity is bioprospecting. That is the search for valuable compounds from wild organisms. This involves searching for, collecting, and deriving genetic material from samples of biodiversity that can be used in commercialized end-products. It has been touted as a mechanism for both discovering new pharmaceutical products and saving endangered ecosystems via the financing of conservation. For example, Rausser and Small [80] claimed that the value of protecting certain ecosystems for bioprospecting can be quite high. Given that the annual market size for products based on genetic resources has been estimated to lie within the range of $220–300 billion [81], there would appear to be strong enough private incentives for mapping, supporting, and protecting diversity through bioprospecting. It should be stressed though that these valuations are not free from criticism. In a seminal paper by Simpson et al. [82] both the marginal benefits and costs of conservation are addressed from a bioprospecting angle. Brock and Xepapadeas [83] emphasized that even one single species can bring potentially very large commercial gains if the marker is large. The increase in the total value of an additional species is therefore considered. The marginal value is, however, decreasing due to redundancy [84]. There are many species that may perform the same function. The total economic cost of losing some species is also included, which is found to be negligible for both high and low levels of species. This, of course, creates a problem for supporting conservation for these compounds unless the case for their contribution is very clear. Contracts have been negotiated between pharmaceutical firms and the government or individuals who control biodiverse ecosystems, although the numbers of private partnerships remain small. For instance, the National Biodiversity Institute (INBio) of Costa Rica negotiated a contract with Merck, in 1989 [85]. This figure is based on sales on world markets for products sold in the healthcare (e.g., pharmaceuticals, cosmetics), agriculture (e.g., seeds, crop protection), and ‘other biotechnology’ (e.g., bioenergy) sectors. In general, however, there is relatively little empirical information regarding these transactions [81].

With these considerations, biopreservation becomes an issue closely linked with the societal-economical aspects of the drug discovery potential of the North-West Pacific, but relevant to the global bio- and chemodiversity as well. Indeed, the current rate of extinction, depending on the taxa, varies from five to 150 species per day—the truly alarming rate, which suggests that the societal-economic potential we were discussing above may be slipping through our fingers in these exact days of the sixth mass extinction—the Pleistocene extinction [5,86]. The importance and urgency of deep-sea biopreservation is currently emerging as the deep-sea mining activities start gaining momentum [87,88].

The Russian Pacific contains several marine protected areas, of which the biggest and most famous is the Far Eastern Marine Biosphere State Nature Reserve, governed by A.V. Zhirmunsky National Scientific Center of Marine Biology, Vladivostok [89]. Created in 1978, it represents the richest studied biodiversity in Russian waters and has obtained the status of a UNESCO international biosphere reserve. The marine reserve occupies 10% of the Peter the Great Bay of the Sea of Japan and includes 11 islands. Its overall area of 643 km^2^ including 630 km^2^ of water is divided into four districts of a varying degree of preservation. The Eastern district has the highest degree of protection with a full ban of entry, collection, or introduction of any organisms; the South and Western districts allow scientific studies; the Northern district is an area where public excursions are permitted. The unique biodiversity of the Far Eastern Marine Reserve has been created through the interaction of the boreal-arctic and subtropical faunas and currently includes 5649 described species, belonging to 38 phyla of six regna (Table 4) [90].

## 6. Recommendations to Unlock the Drug Discovery Potential of the Russian Pacific

Here, we would like to concentrate on the measures to be recommended to unlock the drug discovery potential of the North-West Pacific. These measures can be summarized as part of the national strategic initiative “Marine Natural Products in Health and Disease”.

Russia with its 40,000 km coastal line of 12 seas belonging to three oceans, its extensive marine fleet with the centuries-long tradition of marine expeditions throughout the globe, and its solid scientific tradition of marine research is ideally positioned to become an international leader in the quest to bring marine natural products to the service of humankind. The Russian Far East, being the region of intersection of different marine climatic and ecological zones and the long-standing history of natural products and marine research, appears in this regard as the natural leader in this quest. The national strategic initiative “Marine Natural Products in Health and Disease” builds upon these ideas and needs. We considered that this initiative should be built from the following major blocks:Regular, multi-vessel marine expeditions, both local (at the seas/oceans of the Russian coastline) and global (far away from the Russian coasts), aiming at the massive collection of diverse marine organisms targeted at subsequent extraction of the natural products, accompanied by the proper microbiological, zoological, algal, etc. investigations. Accompanying molecular biology investigations, aiming at the possibility of medium-quality genomic sequencing, appear to be a useful additional angle to these studies. The ultimate goal of this block of the strategic initiative is the creation and constant enlargement of the massive bank of marine extracts, each entry in which is to be accompanied by extensive biological characterization, live specimen preservation where applicable (e.g., in the case of marine microbial collections), and genomic sequencing data.High-throughput multi-scale separation of these extracts, with characterization, maximally permitted for the high-throughput separation, of the resulting mixtures of natural compounds. The ultimate goal of this block of activities is the establishment of a huge collection of fractions and individual natural products in the high-throughput screening-ready format. The scale of this collection should be tens of millions of entries.Establishment of a versatile panel of primary screening assays, all of the high-throughput (HTS) format, aiming at identification of biologically active compounds relevant for the broadest range of medical needs: cancer in its various forms, the broad scope of cardiovascular conditions, the broad scope of neurological diseases, the broad scope of infection diseases, etc. Next, continuous and parallel application of these HTS assays to the screening of the primary marine extracts from block 1 and especially of the fraction/compounds collection of block 2. On the regular basis, new acquisitions to the collections are to be rescreened across the whole panel of the primary assays. The ultimate goals of this block of the national program are two-fold. First, establishment and enlargement of the primary screening assays for various health and disease conditions. Second, generation of the primary hits, originating from the marine natural products, as the starting points for the drug discovery programs (further blocks of the national program) against the respective disease conditions.The national drug discovery/drug development (DD/DD) program based on the marine natural products will start from these primary hits of the previous block. This DD/DD program will have all the features of the typical big pharma DD/DD programs, but will additionally have certain peculiarities, stemming from the specifics of the source of material. These peculiarities are the re-iterative nature of the whole national initiative “Marine Natural Products in Health and Disease”. Specifically, more marine expeditions may be needed to replenish the source of a particular marine organism for the continuous production of the necessary bioactives. These will be complemented, where applicable, with aquaculturing/microbial culturing of the organisms; with genetic production of the bioactive compounds in artificial hosts (which will require the genomic investigations of block 1 of the biosynthetic routes of the bioactives); and with extensive organic synthesis. Otherwise, the typical hit-to-lead optimization, medicinal chemistry, animal experimentation, and clinical studies will be part of this national marine-based DD/DD program. The ultimate goal of this program will be to continuously deliver to the clinical trials and the market of dozens to hundreds of first-in-class or novel drugs against various disease conditions.

Overall, the national strategic initiative “Marine Natural Products in Health and Disease” will coordinate R&D activities across Russia and will involve support and development of the centers of expertise in the following domains:-Marine expeditions and sample collection, including deep-diving expertise.-Marine zoology, microbiology, algal biology, marine ecology; and other aspects of marine biology.-Marine molecular biology, sequencing, genomics, and metagenomics, including big data generation and operations.-Intensive separation and analytics techniques, including massive creation and storage of sample collections.-Massive establishment of HTS-ready primary screening assays for the broadest range of health and disease conditions.-Massive organic synthesis of marine compounds and derivatives.-Marine microbiology and synthetic biology (biotechnological-scale production of marine natural products in natural and engineered hosts).-HTS, drug discovery, and drug development centers of competence on the national scale, possessing capacities all the way from the HTS to clinical trials.

It is expected that the national strategic initiative “Marine Natural Products in Health and Disease” will boost many domains of R&D in Russia, will help solving multiple medical and demographic issues of the country in the long-run, will stimulate economic development of the country, and will position Russia as a global leader in the new branch of technological development of the global importance.

In the context of anticancer drug discovery stemming from invertebrate-derived marine products of the Russian Pacific, especially in comparison with the current activities of the research centers in the region, the following points appear to be of the utmost importance to be developed. First, the focus of the current anticancer screening efforts should shift from the currently dominated superficial analysis of cytotoxic activities against selected cancer lines in vitro to the hunt for candidates for targeted therapies. Indeed, although cancer has been the target of extensive screenings of natural products as the source of drugs, these efforts have so far mostly resulted in appearance of chemotherapeutic agents possessing general cytotoxicity. Such agents more strongly affect metabolically active and proliferating tumor cells, but are also toxic for the healthy cells of the patient. In contrast, development of targeted therapies is the new focus of anticancer research. Such targeted drugs are selective for the cancer cells and have less or (ideally) zero toxicity against healthy tissues as they affect cellular mechanisms, such as oncogenic signaling pathways, which are selectively operational in cancer cells and are instrumental for the proliferation, survival, metastases, and immune control evasion by the tumor. An example of such oncogenic mechanisms is the Wnt signaling pathway, selectively upregulated in several cancer types and especially important in cancer stem cells—the source of tumor recurrence after and resistance against therapy [91]. An HTS platform to screen for inhibitors of the Wnt pathway in cancer cells has been established [92] and applied by us to North-West Pacific marine invertebrates [52]. In this example, collection and characterization of invertebrates during the SokhoBio I expedition to the Kuril Basin was followed by preparation of extracts from these specimen and their screening in the HTS assays, yielding a number of promising hits including anti-Wnt anti-breast cancer activities from *Ophiura irrorata* and other Pacific brittle stars [52]. Upscaling of this approach should be performed to cover many other cancer-specific pathways and mechanisms in an HTS manner. Further, miniaturization and simplification of the HTS assays should be aimed at in order to make them usable directly on the research vessel during the sea expeditions. Such ‘field’ HTS will permit immediate recollection of the hit specimens, partially solving the supply problem of the marine invertebrate drug discovery research [93].

Secondly, massive incorporation of genomics and metagenomics principles into the current marine-based drug discovery programs is urgently needed. These principles will improve the species identification but will also provide other approaches to the solution of the supply problem, by offering the methods of synthetic biology to the current aquaculture methodologies in order to reproduce, in a controlled environment, the desired bioactive molecules. Being successfully applied to other marine regions (see e.g., [94]), these approaches are lagging behind other developments regarding the North-West Pacific resources.

Finally, we wish to stress the fact that the exploration, however limited, of the drug discovery potential of marine invertebrates of the Russian Pacific, has so far been largely restricted to only a few taxonomic groups. Indeed, a mere comparison of Table 2 and Table 4 will show that, of >20 animal phyla and many more smaller taxonomic groups present in the region, only a few (such as star fishes, holothurians, sponges, or ascidians) have been successfully explored in terms of the natural compounds with anticancer properties. Of the taxonomic groups explored, some are well amenable to aquaculturing. Some species of holothurians, for example, are subject to extensive cultivation due to the high gastronomic demand—in the Russian Far East this is the case for the sea cucumber *Apostichopus japonicus* [95,96]. As discussed above, this edible sea cucumber is the source of holotoxin A1 with interesting anticancer properties [50,51], which further adds to the label of *A. japonicus* and several other holothurians as functional foods [97,98,99]. The vast expertise accumulated in the cultivation of holothurians can be applied to new species of this class, which may emerge as the source of promising compounds.

However, more invertebrate groups remain unexplored. This brings us back to the issues discussed in Section 2: Many invertebrate groups, especially those contributing to the meiobenthos, have essentially not been studied even in terms of their species diversity, let alone their drug discovery potential. Untapping this dimension of pharmacophore diversity will require miniaturization of collection, extraction, and HTS approaches.

Clearly, all the other aspects of the national strategic initiative “Marine Natural Products in Health and Disease” listed above will also need parallel developments, but those three discussed here—HTS assays for targeted therapies, genomics/metagenomics approaches, and an extra focus on understudied taxonomic groups—appear as the three currently most underdeveloped.

## 7. Conclusions

In this review, we have attempted to provide an unusual analysis of the anticancer compounds hidden by invertebrates of a particular marine region. Cutting through the issues of biodiversity, chemodiversity, and anticancer pharmacophore diversity of invertebrates of Russian Pacific, we went on to assess the economic and societal benefits that are potentially offered by the development of this treasury. Further, we proposed some strategic measures to be installed, on the national level, in order to convert the enormous drug discovery potential of this marine region into the societal-economic outcome. Clearly, our estimates—on the number of species, of pharmacophores, and of dollars/rubles of potential revenues—were preliminary, yet they were based on the available knowledge and projection principles. We hope that our work, written using a diversity of expertise, will be useful for diverse readerships, such as marine biologists, natural product chemists, drug discovery specialists, as well as politicians, regional and national officials, and strategists. We also hope that our transversal approach to the issue of marine invertebrate-based anticancer drug discovery can be applied to other regions and disease conditions, as well as up-scaled to global dimensions.

## Figures and Tables

**Table 1 marinedrugs-17-00474-t001:** Dynamics of the number of known invertebrate species of in the Russian Pacific.

Source	Number of Species			Total
	Sea of Japan	Sea of Okhotsk	Bering Sea	
Zenkevich, 1963	ca. 2000	ca. 2100	ca. 1500	ca. 3000
Sirenko, 1994	2885	2641	1984	5846
Sirenko, 2013	4077	2798	2414	8411

**Table 2 marinedrugs-17-00474-t002:** Taxonomy of marine invertebrates containing bioactive compounds with experimentally confirmed anticancer activity (classification is presented in accordance with the World Register of Marine Species).

Species *	Bioactive Compounds	Sample Collection Area
**Phylum Echinodermata** **Class Asteroidea** **Order Forcipulatida** **Family Asteriidae**		
*Lethasterias fusca*	Steroid glycosides: Lethasteriosides A and B, thornasteroside A, anasteroside A, and luidiaquinoside	Specimens collected in Aug. 2002 at Posyet Bay (Peter the Great Bay, Sea of Japan), depth 5–10 m; voucher deposited at A.V. Zhirmunsky National Scientific Center of Marine Biology [40].
*Leptasterias ochotensis*	Steroid glycosides: Leptasteriosides A, B, C, D, E, and F, and leptaochotensosides A, B, and C	Specimens collected in Aug. 2003 near Bolshoy Shantar Island (Sea of Okhotsk), depth 20–40 m, “Academician Oparin” 29th scientific cruise; voucher (no. 029-052) deposited at G.B. Elyakov Pacific Institute of Bioorganic Chemistry [41,42].
**Order Valvatida** **Family Goniasteridae**		
*Hippasteria phrygiana (Hippasteria kurilensis)*	Steroid glycosides: Hippasteriosides A, D, C, and D	Specimens collected in July 2003 near Matua Island (Kuril Islands, Sea of Okhotsk), depth 100 m, “Academician Oparin” 29th scientific cruise; voucher (No. 029-26) deposited at G.B. Elyakov Pacific Institute of Bioorganic Chemistry [43].
**Class Holothuroidea** **Order Dendrochirotida** **Family Cucumariidae**		
*Cucumaria fallax*	Triterpene glycoside: Fallaxoside D_1_	Specimens collected in 2011 near Black Brothers Islands (Kuril Islands, Sea of Okhotsk), “Academician Oparin” 41st scientific cruise [44].
*Cucumaria frondosa japonica (Cucumaria japonica)*	Triterpene glycoside: Cucumarioside A_2_–2	Specimens collected in Troitsa Bay (Peter the Great Bay, Sea of Japan) [45].
*Cucumaria okhotensis*	Triterpene glycoside: Frondoside A	Specimens collected in September 2001 in the Sea of Okhotsk near the Western shore of Kamchatka (52°51′00′′ N, 155°56′40′′ E), depth 28 m, by an industrial creep from the small seine-net fishing vessel *MRS-*268 [46].
**Family Sclerodactylidae**		
*Eupentacta fraudatrix*	Triterpene glycosides: Cucumariosides A_1_, A_3_, A_4_, A_5_, A_6_, A_12_, A_15_ and D	Specimens collected in Sept. 1989 in Troitsa Bay (Peter the Great Gulf, Sea of Japan), depth 1–1.5 m; voucher deposited at G.B. Elyakov Pacific Institute of Bioorganic Chemistry [47,48].
**Family Psolidae**		
*Psolus fabricii*	Triterpene glycosides: Psolusoside А, B, C_1_, C_2_ and D_1_, B, E, F, G, H, H_1_, and I	Specimens collected in Aug.–Sept. 1982 near Onekotan Island (Kuril Islands, Sea of Okhotsk), depth of 100 m, by fishing seiners “Mekhanik Zhukov” and “Dalarik”, identified by Prof. V.S. Levin; voucher specimens deposited at A.V. Zhirmunsky National Scientific Center of Marine Biology [49].
**Order Synallactida** **Family Stichopodida**		
*Apostichopus japonicus* (*Stichopus japonicus*)	Triterpene glycoside: Holotoxin A1	Specimens collected in Posiet Bay (Peter the Great Bay, Sea of Japan) [50,51].
**Class Ophiuroidea** **** **Order Ophiurida** **** **Family Ophiuridae**		
*Ophiura (Ophiuroglypha) irrorata (Ophiura irrorata)*	Methanol extracts	Specimens collected in June–Aug. 2015 in the Kuril Basin of the Sea of Okhotsk, depths 1700–4750 m, “Academician Lavrentyev” SokhoBio expedition [52]
**Phylum Porifera** **Class Demospongiae** **Order Poecilosclerida** **Family Crambeidae**		
*Monanchora pulchra*	Alkaloids: Monanchocidins A and B, monanchomycalins B and C, ptilomycalin A, normonanchocidin D, urupocidin A, and pulchranin A	Specimens collected in Aug. 2008 near Urup Island (46°07,0 N; 150º02,1 E, Sea of Okhotsk), depth 150–175 m, “Academician Oparin” 36th scientific cruise [53,54,55].
**Family Myxillidae**		
*Melonanchora kobjakovae*	Fatty acids: Melonosides A and B, melonosins A and B	Specimens collected in July 2011 near Urup Island (46°02,1 N; 149°55,3 E), depth 121 m, “Academician Oparin” 41st scientific cruise; voucher (No. PIBOC O41-135) deposited at G.B. Elyakov Pacific Institute of Bioorganic Chemistry [56,57].
**Phylum Chordata** **Subphylum Tunicata** **Class Ascidiacea** **Order Aplousobranchia** **Family Didemnidae**		
*Polysincraton sp.*	Polyketides: Mycalamide A	Specimens collected in Aug. 2008 in Natalyi Bay off Urup Island (Kuril Islands, Sea of Okhotsk, 46°18′30′′N, 150°15′30′′E), depth 166–200 m, “Academician Oparin” 36th scientific cruise; voucher deposited at G.B. Elyakov Pacific Institute of Bioorganic Chemistry [58].

* Taxa in parentheses are synonymous.

**Table 3 marinedrugs-17-00474-t003:** Some examples of marine compounds form the North-West Pacific with brief description of their anticancer effects.

Structure	Name	Anticancer Activities in Vitro	Anticancer Activities in Vivo	Ref.
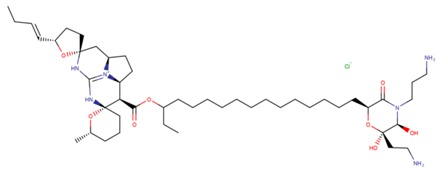	Monanchocidin A	-Cytotoxicity (leukemia, cervix carcinoma, epidermal, genitourinary cancer, prostate cancer, bladder cancer).-Synergism with cisplatin.-Autophagy, cell cycle arrest, apoptosis.-Inhibition of EGF-induced neoplastic transformation.	N/A	[53,54,55]
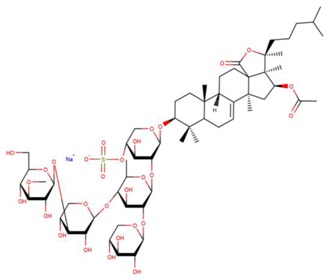	Frondoside A	-Cytotoxicity (prostate cancer, EGF-driven epidermal, cervix carcinoma, leukemia, urothelial carcinoma).-Inhibition of colony formation (prostate cancer).-Cell cycle arrest, apoptosis.-Inhibition of membrane transport of P-glycoprotein (multidrug resistance).-Synergism with cisplatin and gemcitabine.	-Mouse xenograft model (2 prostate cancer lines): suppression of tumor growth, circulating tumor cells, and lung metastasis.-No side effects. Immune modulating response.	[46,59,60,61,62,63]
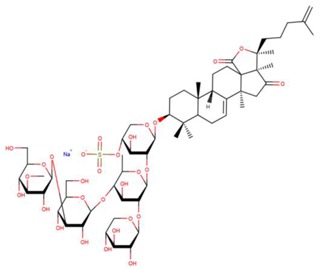	Cucumarioside A2-2	-Inhibition of proliferation (EGF-driven epidermal, cervix carcinoma).-Inhibition of colony formation.-Inhibition of membrane transport of P-glycoprotein (multidrug resistance).	N/A	[61,62]
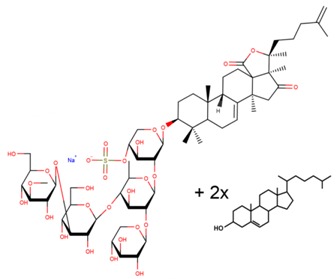	Cumaside	-Cytotoxic properties significantly reduced as compared to parental cucumarioside A2-2.-Immuno-modulatory effects.	-Antitumor effects in mouse Ehrlich carcinoma models.-Immuno-modulatory effects.	[33,64,65]
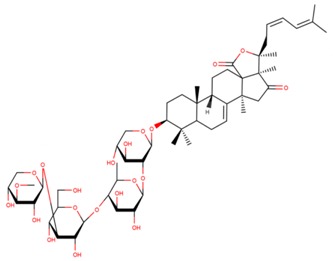	cucumarioside A5	-Cytotoxicity (spleen lymphocytes, Ehrlich carcinoma).	N/A	[47,48]
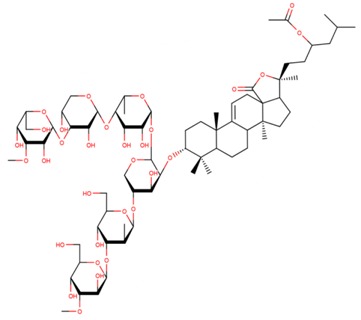	Holotoxin A1	-Apoptosis (leukemia, colorectal cancer).-Activation of acid sphingomyelinase and neutral sphingomyelinase.	N/A	[50,51]
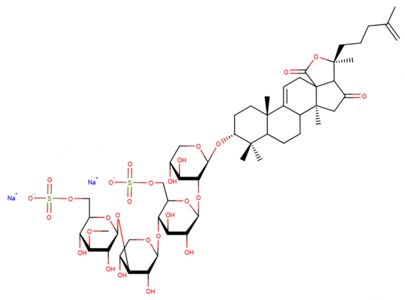	Psolusoside А	-Inhibition of proliferation and colony formation (EGF-driver epidermal).	N/A	[49,66,67,68]
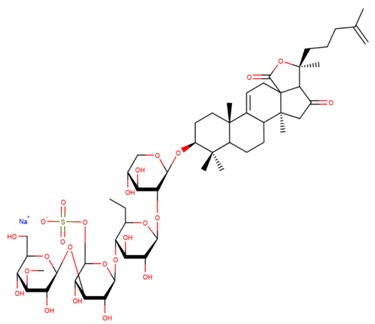	Psolusoside E	-Inhibition of proliferation (neuroblasoma).-Inhibition of colony formation (colorectal adenocarcinoma).	N/A	[69]
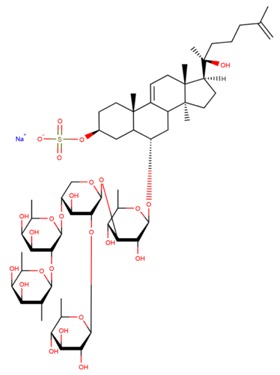	Leptasterioside B	-Inhibition of proliferation and colony formation (melanoma and breast cancer).	N/A	[41]
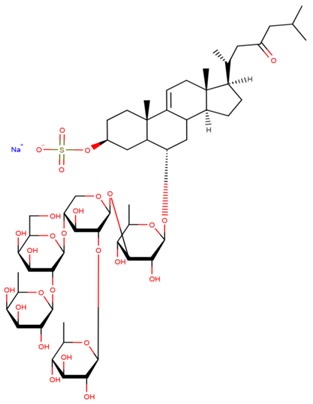	Lethasterioside A	-Inhibition of colony formation (melanoma, breast cancer, colorectal carcinoma).	N/A	[40]
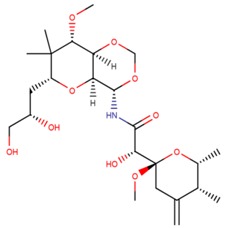	Mycalamide A	-Cytotoxicity, inhibition of neoplastic transformation (EGF-driven epidermal).-Inhibition of colony formation (epidermal and cervix carcinoma).-Inhibition of AP-1- and NF-κB- transcription.	N/A	[58]
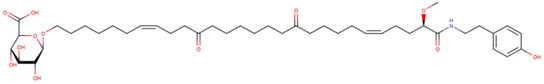	Melonoside A	-Autophagy (germinal tumor).-Inhibition of AP-1- and NF-κB-transcription.	N/A	[56,57]

**Table 4 marinedrugs-17-00474-t004:** Biota of the Far Eastern Marine Reserve.

Phylum	Number of Species
Superregnum Prokaryota Regnum Eubacteria	
Cyanobacteria	217
Superregnum Eukaryota Regnum Protozoa	
Euglenozoa	109
Regnum Plantae	
Bryophyta	76
Charophyta	212
Chlorophyta	314
Glaucophyta	1
Marchantiophyta	45
Rhodophyta	81
Tracheophyta	904
Regnum Fungi	
Ascomycota	466
Basidiomycota	66
Regnum Chromista	
Bacillariophyta	522
Cercozoa	1
Cryptophyta	11
Foraminifera	78
Haptophyta	1
Myzozoa	151
Regnum Animalia	
Annelida	248
Arthropoda	825
Brachiopoda	1
Bryozoa	16
Cephalorhyncha	1
Chaetognatha	5
Chordata	528
Cnidaria	41
Ctenophora	4
Echinodermata	38
Mollusca	340
Nematoda	121
Nemertea	22
Phoronida	2
Platyhelminthes	12
Porifera	3
Rotifera	14
Sipuncula	3
Tardigrada	1
Xenacoelomorpha	6
Total	5649

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
