# Peer review of "The Anticancer Drug Discovery Potential of Marine Invertebrates from Russian Pacific"

_marinedrugs, 2019, doi:10.3390/md17080474_

Round 1

Reviewer 1 Report

No comments 

Reviewer 2 Report

 I found that a substantial improvement was made to the manuscript on
 "The anticancer drug discovery potential of marine invertebrates from Russian Pacific".

 I recommend the manuscript for publication to Marine Drugs.

This manuscript is a resubmission of an earlier submission. The following is a list of the peer review reports and author responses from that submission.

Round 1

Reviewer #1.

  1. This article is of a great interest but for a wider audience of nonspecialized scientists.

We sincerely thank the reviewer for the general assessment that our paper is of a great interest.

We further wish to stress that it has been our goal to broaden the scope of the manuscript in such a manner that it becomes interesting to the readership larger than the community of marine natural product scientists. In addition to this highly respected group of researchers, we wished the paper to attract potential interest from various drug discovery professionals, ecologists, economists, and more. Apparently, this goal of broadening the readership scope has been successful. We would politely disagree if the reviewer might be inclined to suggest that such broadening of the scope is not good for Marine Drugs. We are convinced that such broadening of the readership is great for any journal, certainly including Marine Drugs, whose aims and scope are editorially defined as being “on the discovery, development, exploitation, and production of biologically and therapeutically active compounds from marine habitats”.

  1. Some references are difficult to access and cannot be read by most concerned scientists.

This notion of the reviewer is certainly correct, but it relates to the fact that these exact references are for papers / books either published a long time ago, and /or published in Russian. In this regard, we are actually certain that it is a strength, rather than a weakness of our review article, that we bring to the modern readership’s attention works which are otherwise poorly accessible and of which the modern international scientific community is thus not fully aware.

  1. The link to the clinical trails pipeline is wrong (uncomplete). Would be better in the references.

We thank the reviewer for this notion. To address this issue, we have done the following. First, we have amended the information in the manuscript with the freshly deposited data on the portal http://marinepharmacology.midwestern.edu/clinical_pipeline.html. Second, we searched through various sources to illustrate that more marine-derived drugs have been approved or entered clinical trials. Finally, we put the link to the database into references.

In accordance with these modifications, the respective part of the text now reads as follows (lines 175-183):

Alejandro Mayer from Midwestern University maintains a highly informative web-site tracking clinical development of marine-derived drugs [26]. Currently, this web-site lists 8 such drugs as approved, 6 in phase III, 14 in phase II, and 10 more in phase I clinical trials. Of this total of 38 molecules, 32 are anticancer agents, mostly of the cytotoxic or cytostatic type. This list can be further expanded somewhat. Indeed, in addition to omega-3 acid ethyl esters (Lovaza®), icosapent ethyl (Vascepa®) and omega-3 carboxylic acids (Epanova®) have been approved [29]; keyhole limpet hemocyanin is used as an anticancer vaccine conjugate [30]; linear sulfated polysaccharides from Rhodophyceae seaweeds are applied as an anti-viral agent [31]; or OligoG derived from marine algae is being tested for cystic fibrosis in phase II trials [32].

  1. Many spelling mistakes (bentos, desease, biodiversity...). The abbreviations of the units do not comply with international standards.

We thank the reviewer for having noted these mistakes, which have now been corrected in the revised version of the manuscript.

Reviewer #2.

  1. This paper summarizes biodiversity, chemodiversity, and the anticancer pharmacophore diversity of the marine invertebrates in the North-West Pacific region. The paper is composed of following 5 issues.
  2. Assessing the biodiversity of marine invertebrates of North-West Pacific
  3. Natural products from North-West Pacific invertebrates
  4. Marine-derived compounds from North-West Pacific invertebrates with an anticancer potential
  5. The drug discovery potential of invertebrate compounds of the North-West Pacific
  6. Recommendations to unlock the drug discovery potential of the Russian Pacific

For the issues 1-2, assessment of biodiversity and chemodiversity are solely based on simple assumption and limited data. It would be more interesting to the readers of Marine Drugs if more extensive analysis, based on rich data, was presented.

We are somewhat puzzled by the negative tone of the reviewer regarding the issues on biodiversity and chemodiversity of the North-West Pacific. The goal of these two first sections of our review article are to i) review the available information on the bio- and chemodiversity of this marine region, and ii) to provide estimates, based on the dynamics of species and chemospecies discovery, of the overall diversity in the biological and chemical spaces this region has to offer. While Marine Drugs and other marine-dedicated journals often deal with particular marine regions, to our knowledge the North-West Pacific has not been covered by any recent reviews, in the sense of its bio- and chemodiversity. Yet this is a huge region with its peculiar characteristics and often unique species. Thus, we are sure that the review of the available data for this region represents a great interest by itself. The fact that the data available are somewhat limited is unquestionable, but this limitation applies of course to most other marine regions, making it actually even more interesting to review the available data.

In a way partially addressing the issue of data limitation, we have now provided Table 2 with the details on species collection behind the isolation of compounds with anticancer properties, as well as broader discussion on biopreservation with information on a deep-sea Far Eastern Marine Reserve, representing the richest studied biodiversity in Russian waters and having the status of a UNESCO international biosphere reserve.

Regarding the simplicity of assumption – here the Reviewer probably refers to the prognosis on the overall diversity we make based on the available data and on the dynamics of acquisition of new data. We did our best, and we doubt that any better prognosis could be made in this regard.

  1. For the issue 3, it would be more interesting and informative if more extensive and detailed summary was presented.

Following this advice of the reviewer, we have significantly expanded this section. First, more detailed descriptions of the anticancer activities of the compounds have been provided. Second, more compounds originating from North-West Pacific are now covered in the revised manuscript as examples. Third, we have added two additional tables providing details on species collection behind the isolation of compounds with anticancer properties (Table 2) and on structures of representative compounds (Table 3).

  1. Regarding the issue 4, only the general discussions on the potential of invertebrate compounds are reiterated, and fails to suggest new insight on this issue.

We would politely disagree with the Reviewer’s assessment of this section of our review. A plenty of review articles on natural products possessing these or those biological activities have been published. Hundreds of natural products scientists perform academic studies on the natural compounds. However, there is an unquestionable gap between these academic studies and the drug discovery/development efforts of pharma industry. As the Reviewer is certainly aware, only a tiny fraction of interesting molecules possesses properties needed to enter the drug development pipeline, with the majority of those being further eliminated along the path to the drug.

These are the reasons for us to include a section discussing the drug discovery potential of natural compounds of the North-West Pacific. In our eyes, as well as apparently in the eyes of two other Reviewers, this angle of analysis is a strong add-on to what is typically being reviewed in natural products articles. Our own drug discovery/development experience has helped in drafting this section. We hope that this addition will expand the audience of the readership of this article beyond the field of marine natural products, including, for example, drug discovery industrials.

  1. Regarding the issue 5, it sounds like a proposal or suggestions to Russian government. It may be difficult to draw attention of general readers.

Once again, we would politely disagree with this position of the Reviewer, for the following two reasons. First, Russian sea waters are huge and to various degrees unique. In this regard, discussion of the measures needed to fully explore the marine drug discovery potential of these waters is of an obvious interest, no less than discussing such potential of e.g. US, British, or North Caledonian waters, which can be seen in different international journals. Second, planning of national programs aiming at the full-scale exploration of this potential is interesting for readers from other countries which may be involved or willing to become involved in similar national programs. Finally, these national efforts certainly are and will in future even more be integrated in the multinational global programs.

  1. The authors stated in the conclusion, i.e. “Clearly, our estimates – on the number of species, of pharmacophores, and of dollars/rubles of potential revenues – are preliminary, yet they are based on the available knowledge and projection principles.” It may be better if this review includes more valuable information rather than preliminary estimations. This review contains simply general and well-established cognition on biodiversity and chemodiversity. And the discussion on marine-derived compounds from North-West Pacific invertebrates with an anticancer potential fails to deliver new insight or valuable information.

We wish to thank the Reviewer for his/her detailed analysis of our manuscript. The issues mentioned in this point have been addressed in our responses to the points above. We wish to reiterate that we have aimed at writing a review article going beyond the standard overview of bioactive compounds. We respect the opinion of the Reviewer and of some prospective readers that such “going beyond” may appear, at least on the first glance, as discomforting. Yet we are sure that many other readers, and perhaps this Reviewer upon the second reading, will appreciate the value of our intentional broadening of the scope of the article. We also wish to stress again that we have, in this revised version, provided significantly more details on the compounds with anticancer properties, following the demand of the Reviewer.

Reviewer #3.

  1. The study entitled "The anticancer drug discovery potential of marine invertebrates from Russian Pacific" is a very much needed effort where it has brought into the attention of marine biologists towards a region rich in biodiversity. Indeed the work is of good scope to the wide range of readers.

We wish to sincerely thank the Reviewer for this positive assessment of our manuscript.

  1. However, I would recommend the authors to highlight the findings in the form of a table where all the discovered compounds from North-West Pacific invertebrates should be enlisted according to their potential or chemical nature.

We thank the Reviewer for this suggestion; such a table has now been added (Table 3).

  1. Furthermore, authors also addressed to focus the biodiversity preservation which is a valid point, however, it would be more appropriate if authors consider their literature search again and express which oceanic regions should be dealt with much higher attention and care in this regard.

We thank the Reviewer for this suggestion, in accordance with which we have now significantly expanded the respective section. Specifically, considerations and relevant very recent references regarding deep-sea biopreservation have been added, along with overview of the UNESCO international biosphere reserve located in this region (lines 477-491 plus Table 4).

  1. A third and most important point which is missing from this work is the list of species/genera which have already been researched for such compounds and the species/genera which have still not been explored and are potential candidates.

The Reviewer raises two important and interconnected issues here. First, we have now added a table (Table 2) listing the species and details on their collection location, depth, etc, which illustrates the scope of the taxonomic groups which have been investigated in regard with compounds with anticancer properties. This Table 2 also serves as the feed to the Table 3 which in turn shows the chemical species and their activities in the anticancer aspect.

Second, and more importantly, the Reviewer asks about which taxonomic groups of the region have not yet been properly analyzed in regard with the anticancer potential. To address this important issue, we have now added a paragraph to the respective section (lines 600-617) and stressed the importance of exploration of these poorly studied groups for their drug discovery potential.

  1. Are there some potential species in this region that should be farmed and could be considered as functional foods in this regard?

Although this issue goes beyond the main topic of our review, we have now added some considerations on it, in a link with the questions on aquaculturing (pp. 605-612).